# Understanding opposing predictions of *Prochlorococcus* in a changing climate

Vincent Bian[1], Merrick Cai[2] & Christopher L. Follett [3] ✉

Statistically derived species distribution models (SDMs) are increasingly used to predict ecological changes on a warming planet. For *Prochlorococcus*, the most abundant phytoplankton, an established statistical prediction conflicts with dynamical models as they predict large, opposite, changes in abundance. We probe the SDM at various spatial-temporal scales, showing that light and temperature fail to explain both temporal fluctuations and sharp spatial transitions. Strong correlations between changes in temperature and population emerge only at very large spatial scales, as transects pass through transitions between regions of high and low abundance. Furthermore, a two-state model based on a temperature threshold matches the original SDM in the surface ocean. We conclude that the original SDM has little power to predict changes when *Prochlorococcus* is already abundant, which resolves the conflict with dynamical models. Our conclusion suggests that SDMs should prove efficacy across multiple spatial-temporal scales before being trusted in a changing ocean.

Plankton are involved in nearly every fundamental biogeochemical process in the oceans, feeding global fisheries production and driving the marine carbon cycle[1–3]. Microbial populations are in turn supported by nutrient supplies, and their growth rates modified by light and temperature[4,5]. Since microorganisms are directly affected by, and in turn directly affect their environment, it is crucial to understand the impact that physical and chemical factors have on these populations[6]. Great progress has been made both through the generation of prognostic dynamical models[7–10] and through statistical data-driven approaches[11–15].

Time dependent, differential equation based, population dynamics models provide one method to explore what drives microbial populations in the sea. Most models of this class resolve only a few plankton types[16,17], but our capabilities for modeling a diversity of plankton groups has greatly increased[10,18,19]. In general, these models predict that the total global concentration of phytoplankton biomass in the surface ocean will decrease with warming[20,21], with localized increases in high latitude regions where nutrients are more plentiful and changes in light and temperature have a larger impact on growth[10,17]. Mixing processes bring deeper, nutrient laden waters to the surface where they support vigorous plankton growth. As the surface ocean warms, the thermal gradients (stratification) in the surface ocean strengthen. This decreases vertical mixing and the nutrient supply for phytoplankton growth. In the ocean's gyre regions, where small picoplankton are already a large fraction of the biomass, this decrease in nutrient supply can lead directly to a decrease in the biomass of small cells[22]. When growth rates are limited by the supply of nutrients, like in oligotrophic gyres, small plankton have an advantage because of their high surface area to volume ratio[23]. In high latitude regions where nutrients are more plentiful, enhanced stratification from surface warming is thus predicted to increase the abundance of small cells relative to large plankton with decreasing nutrient supply. The range of small phytoplankton is thus expected to increase.

Species distribution models (SDMs) take a complementary approach to population dynamics models and aim to predict the population of a species directly from data using a reduced set of predictors[13,24]. When conditions are right, these models can reliably and accurately predict the population size in different environments, and be extended beyond the data used to parameterize them[25,26]. Correlative SDMs are statistical models based on correlations between

[1]Department of Physics, Massachusetts Institute of Technology, Cambridge, MA, USA. [2]Department of Mathematics, Massachusetts Institute of Technology, Cambridge, MA, USA. [3]Department of Earth, Atmospheric and Planetary Sciences, Massachusetts Institute of Technology, Cambridge, MA, USA. ✉e-mail: follett@mit.edu

the distribution of a species and environmental factors. They are efficient to build and can incorporate all available ancillary data. With an increase in the availability of high quality plankton data, these models have been generated to predict plankton populations and their diversity in a modern and changing ocean[27–31].

Determining the validity of both these model types can be difficult because of the spatial patterning of ocean data[32]. The ocean can be separated into physical and biophysical provinces with sharp spatial transitions[33,34]. This, combined with the nonlinear nature of ecosystem population dynamics, suggests distinct population regimes in the sea[35–37]. Differences between model predictions and measurements can thus be thought about in terms of 'pattern errors' and 'magnitude errors'[38]. Differences can be caused by the shifting of regime boundaries in space, or by the modification of population levels within a province itself[39]. When statistical models are built from global datasets, both pattern and magnitude errors can influence the goodness of fit. Thus, it becomes critical to understand why a model has a good fit in order to determine under which circumstances its predictions should be trusted.

Here, we consider the plankton prediction problem in the context of surface ocean (depth < 50 meters) populations of the globally dominant phytoplankton *Prochlorococcus*[40–43]. Discovered in 1988[40], *Prochlorococcus* resides primarily between 40° N and 40° S, thriving in the well lit surface waters. Due to its small size, *Prochlorococcus* dominates low-nutrient (oligotrophic) areas of the ocean where its high surface area to volume ratio provides an advantage for acquiring nutrients[43]. The abundance of global concentration data for *Prochlorococcus* makes it ideal for constructing statistical, machine learning based SDMs[14] (See schematic in Fig. 1). The importance of both *Prochlorococcus* and the model constructed in Flombaum et al. 2013 make it ideal for exploring the extendability of SDMs for plankton prediction under climate change. Flombaum et al. apply multiple techniques for building correlative SDMs: artificial neural network models, non-parametric models, and a parametric regression[44,45]. For the problem of predicting *Prochlorococcus* abundance, the parametric regression model was not only the simplest, but also the most effective[14]. Based entirely on temperature and photosynthetically active radiation (PAR), the model predicts that *Prochlorococcus*

concentrations increase monotonically with temperature, and with PAR up to a threshold value[14]. This model is combined with output of sea surface temperature changes predicted by earth system models to predict large, systematic increases in *Prochlorococcus* populations by 2100[14,15,46]. These predictions have large implications for topics ranging from understanding future changes in global microbial biodiversity[47,48] to carbon sequestration driven by biological export out of the surface ocean[49–51].

Recent work has extended the model to other plankton types and exposed a fascinating and important conflict[15]. While this statistical model for plankton populations suggests large increases in *Prochlorococcus* and other small plankton in the surface waters of the ocean gyres, global population dynamics simulations suggest the opposite[18–20,22,52,53]. Additionally, recent statistical work on a dataset of *Prochlorococcus* collected from new transects isolated in the subtropics suggests that the temperature sensitivity of SDMs changes sign depending on which ancillary variables are included in the analysis[27]. Thus, the model predictions appear sensitive to both the spatial extent of the dataset, and to which ancillary variables are used. Understanding the underpinnings of such dramatically different predictions among SDMs and population dynamics models is important. As SDMs become more prevalent and are used to make decisions about our future ocean, understanding when they should be trusted is imperative[31]. *Prochlorococcus* is an ideal test case: it is important biogeochemically; large, global datasets exist for it; and a conflict exists between dynamical and statistical model predictions.

As the temperature warms dynamical models predict that the range of small-celled *Prochlorococcus* will expand while its concentration decreases[10,17]. This is due to increased stratification which decreases nutrient supply. Can we build a similar understanding for the predictions of the SDMs? Unfortunately, understanding the predictive power of SDMs can be difficult[54,55]. While fitting a model to global datasets, the pattern and magnitude errors must be carefully considered[38]. Temporal forcing and the inclusion of strong forcing axes like depth (phytoplankton do not grow in the dark) may additionally smear observations across parameter space, making continuous models appear valid when they are not. These are some reasons why SDMs trained on modern simulated data have difficulty

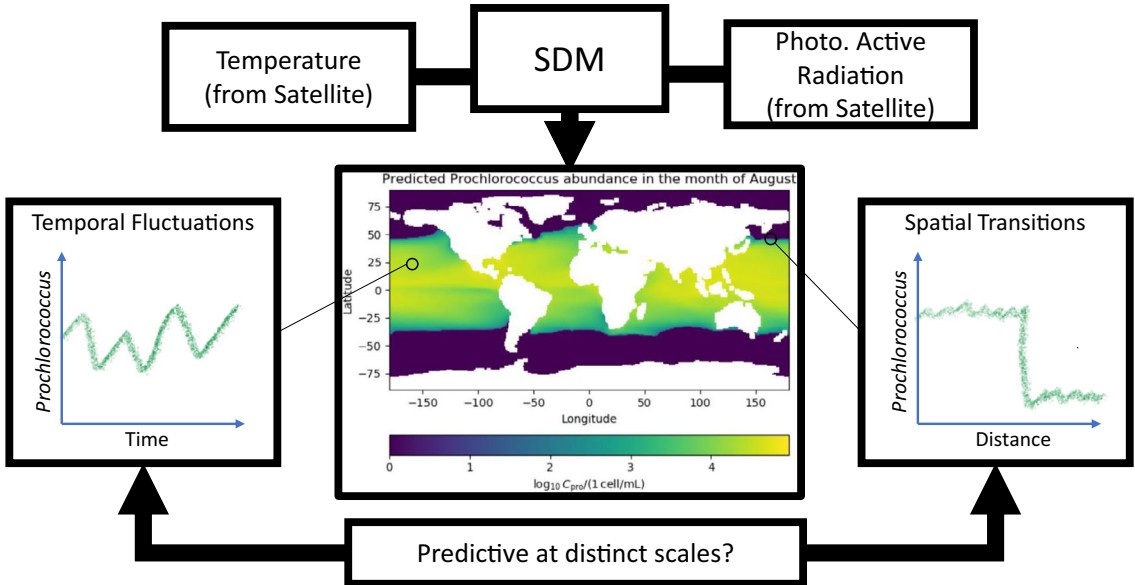

**Fig. 1 | Schematic of the operation of a species distribution model.** SDMs, like the Flombaum model[14,15], take observed variables and provide predictions for species abundance. The Flombaum model uses temperature and light (Photosynthetically Active Radiation, PAR) in an SDM to predict the concentration of *Prochlorococcus* cells in the ocean. The schematic shows how this works using satellite data for the climatological month of August. We explore the power of these models at distinct spatial-temporal scales by focusing on local temporal fluctuations and sharp spatial transitions in species abundance.

under simulated warming[55]. Ideally, we would build and test SDMs directly using experiments[56], but often this is impractical. Using field observations, however, we can test whether models and their dependent variables maintain predictive power across multiple, distinct, spatial-temporal scales. If a variable like temperature is predictive in many different regimes, it is more likely that shifting it will lead to predictable changes. Applying this idea, we first focus on population fluctuations about a mean state. When populations are stable, do small changes in the driving variables correlate with changes in abundance? Second, many populations experience sharp spatial transitions between regions of differing abundance[41,57]. Do these transitions cluster systematically when plotted against the dependent variables? These ideas can be combined by looking at the correlation structure of high resolution oceanographic transects as a function of scale.

Although the Flombaum model is statistical[58], we posit that if the population is highly correlated with temperature and light across multiple spatial-temporal scales, then it may generate accurate predictions under future conditions. This could be due either to the direct, causal, relationships between temperature, light and the relative growth rates of the organisms, or due to hidden mechanisms which connect temperature and light to nutrient and physical dynamics[59]. The mechanistic connection does not need to be known for a model to be predictive. We focus on correlations between *Prochlorococcus* populations in the surface ocean, light, and temperature under three situations: global surface data and the predictive power of the Flombaum parametric regression model; the correlations of light and temperature over time using long-term time series data; and the spatial-temporal transitions between regions of high and low population levels (See schematic in Fig. 1). We go on to demonstrate the

connection between the spatial scales of fluctuations in *Prochlorococcus* abundance, temperature, and predictability by analyzing correlations across a continuum of spatial scales. Our results provide additional insight into how and why *Prochlorococcus* populations may shift in the future, and strongly suggest the need for models to demonstrate predictive power across a continuum of scales before being trusted under future conditions.

## Results

The Flombaum model was constructed using a dataset containing data from 103 cruises covering every major ocean basin. The dataset includes colocalized measurements of longitude, latitude, and *Prochlorococcus* abundance as measured by flow cytometry[14,15]. We first reduce the dataset to the ocean's surface, including only data taken at a depth of at most 50 meters that contains coincident PAR and temperature measurements. A direct comparison of the Flombaum model and the surface measured values is shown in Fig. 2a (11930 datapoints). The *Prochlorococcus* abundance forms two main clusters: a set of measurements very close to zero (6568 datapoints), and a more spread out cluster of nonzero measurements (5362 datapoints). To remain consistent with Flombaum et al. 2013, for log-space calculations we have reset zero measurements to 1 or $\log_{10}1 = 0$ in log-space.

The model captures the mean of the main non-zero data cloud. The distinct cluster of near zero measurements, however, appears systematically overestimated with a large range in predicted values. One potential reason for this is that the Flombaum model is less predictive near the edge of the species' spatial range (region from light to dark in Fig. 1). This hierarchical structure in the model fit matches our understanding of the broad biogeographical patterns of

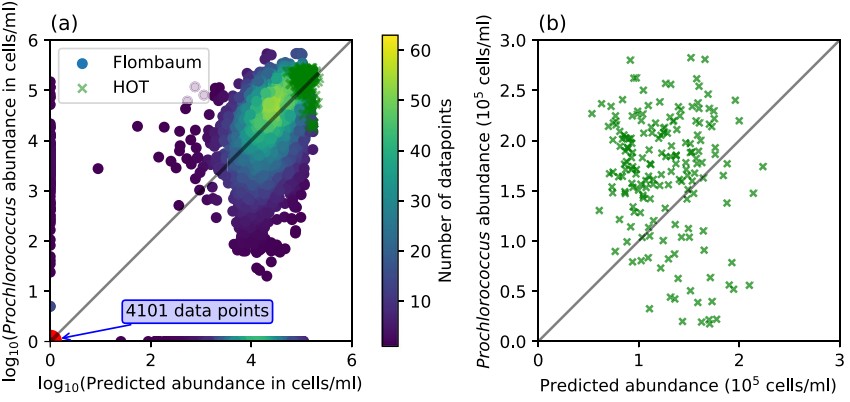

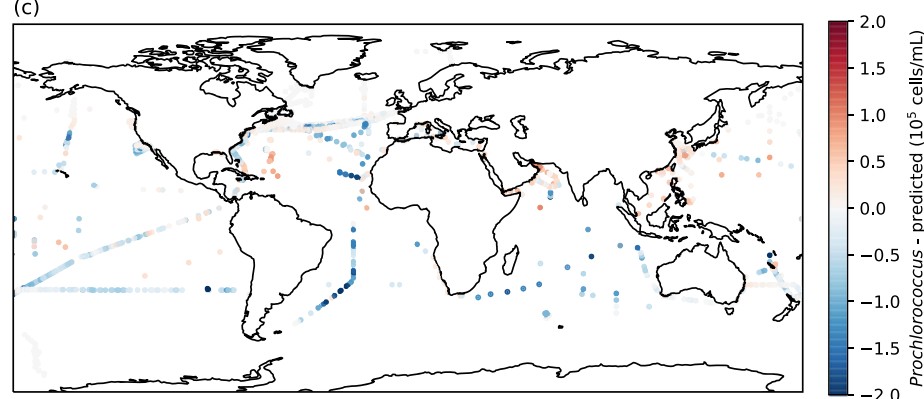

**Fig. 2 | Comparison of model predictions and observations. a** A log-log plot of the *Prochlorococcus* abundance predicted by the Flombaum model, vs the measured abundance, including data points from both the original Flombaum dataset and the HOT dataset. **b** A linear scale plot of the predicted vs actual *Prochlorococcus* abundance for surface data from Station ALOHA. **c** A map of surface locations, colored circles, within the Flombaum dataset. Red indicates an underprediction and blue an overprediction.

*Prochlorococcus*: large regions of relatively constant values and large regions of none. To evaluate the Flombaum model's geographical dependence, we consider the difference between predicted and measured *Prochlorococcus* abundance versus geographic location. The results are shown in panel c of Fig. 2 (the same 11930 datapoints as panel a). Dark blue regions in the North Pacific and South Atlantic (30° N and 30° S) occur in regions known to be near the geographic range of the organism[42]. This supports our assertion that the the strong bi-modality in the prediction of this model may be due to its changing predictive power near spatial transitions[57].

Additionally, there is high variance within the high-concentration cluster, which suggests exploring how the model captures variability over time. We compare Flombaum model predictions with measured data taken at a single location (green crosses in Fig. 2a, b). The Hawaii Ocean Time-series (HOT) contains monthly measurements of *Prochlorococcus* abundance, starting from December 1990, as well as a suite of other measurements including temperature and PAR[48,60]. The result of this comparison is shown in Fig. 2b (183 datapoints), with Station ALOHA located just north of Hawaii in Fig. 2c. At the global scale, acting as a single datapoint, Station ALOHA matches the predictions of the Flombaum model. In the restricted dataset, however, the correlation between prediction and measurement is substantially weaker, suggesting that the Flombaum model is partially confounded by the effects of other variables and processes. Specifically, the main axis of variation in the ALOHA dataset is not aligned with the axis of prediction as shown by the vertically elongated data cloud in Fig. 2b.

This discrepancy is especially clear in Supplementary Fig. 2 where the data is compared directly with temperature and PAR.

It is important to state clearly that the Flombaum model was built as a global scale predictor and it is not clear that it can or should be applied down-scale, either in time or space. The predictive power of the Flombaum model near the boundaries of the *Prochlorococcus* range, and over short time periods, may not reflect the accuracy of global scale predictions of the model, such as how *Prochlorococcus* is expected to proliferate under climate change. However, we expect that temperature and light, the input variables of the model, to remain the driving variables even if the model structure is scale dependent.

### Temporal fluctuations

One way to explore whether light or temperature drive *Prochlorococcus* is to determine how relatively small changes in these variables correlate with changes in abundance. Returning to the Hawaii Ocean Time Series station we compare changes in the monthly temperature and PAR (with depth < 50 meters) with changes in the monthly average abundance of *Prochlorococcus* (See Supplementary Fig. 1). The resulting plots (using the same 183 datapoints as Fig. 2b) are shown in Fig. 3a, b. Contrary to predictions made by the Flombaum model, temperature changes are not positively correlated to *Prochlorococcus* abundance (Pearson's correlation coefficient $R = -0.02 \pm 0.12$). Changes in PAR are only weakly negatively correlated with changes in *Prochlorococcus* ($R = -0.35 \pm 0.12$, $R^2 \approx .12$).

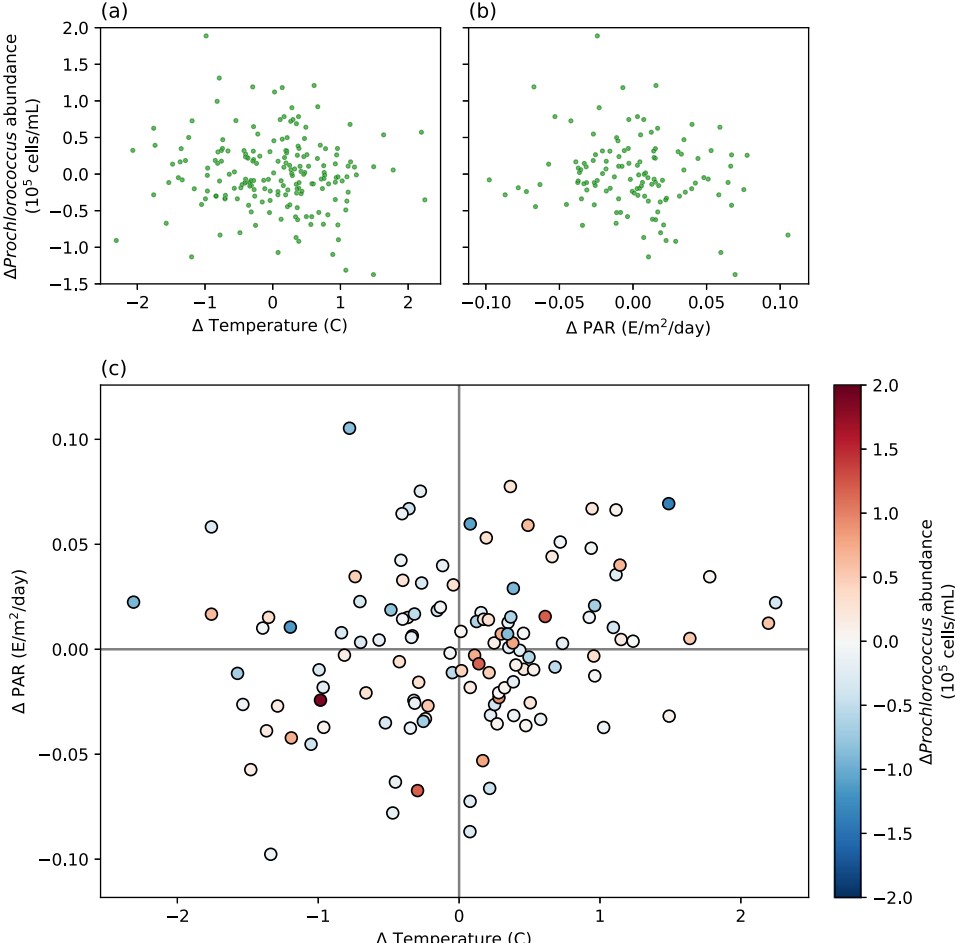

**Fig. 3 | Temporal fluctuations at Station ALOHA do not correlate strongly with light and temperature.** Month to month changes in *Prochlorococcus* population in the upper 50 meters vs changes in temperature (**a**) and Photosynthetically Active Radiation (PAR) (**b**). Changes in PAR vs. temperature vs. *Prochlorococcus* are shown in **c**.

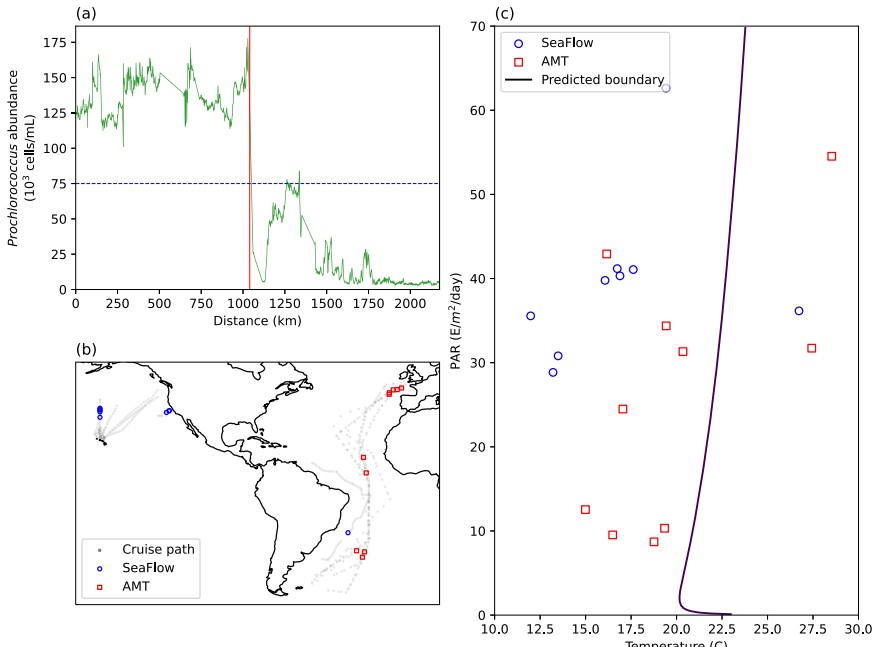

**Fig. 4 | Boundary locations do not follow a contour of light and temperature.** Strong shifts in *Prochlorococcus* concentration along surface transects (such as the one shown in **a**) representing niche transitions are plotted on the map (**b**) and in co-localized PAR and temperature space (**c**). The dark curve is the predicted boundary from the Flombaum model.

However, both temperature and light could also act together to influence the populations of *Prochlorococcus*. We plot the monthly changes in light, temperature, and *Prochlorococcus* together in Fig. 3c. For temperature and PAR regimes contained in the HOT dataset, the Flombaum model predicts that *Prochlororoccus* monotonically increases as a function of temperature, and monotonically decreases as a function of increasing PAR for the range of PAR values found in the surface ocean. Thus, we would expect more increases in the lower-right quadrant of Fig. 3c, and more decreases in the upper-left quadrant. Indeed, $57 \pm 10\%$ of the data points in the lower-right quadrant represent increasing *Prochlorococcus*, while $21 \pm 8\%$ of the data points in the upper-left quadrant represent increasing *Prochlorococcus*, as compared to $44 \pm 5\%$ of the data points in the whole plot. Performing a multivariate correlation analysis with both PAR and temperature yields a combined $R^2 = .125 \pm .04$, suggesting that roughly 12% of the fluctuation in *Prochlorococcus* may be explained simply by fluctuations in light and temperature at this location. This being the same value as the correlation for light alone, however, suggests that there remains little predictive power in temperature fluctuations at the monthly timescale for surface populations.

**Spatial transitions**

Using data collated in the Simons CMAP database[61], we investigated how well temperature and PAR predict locations separating regions of high and low *Prochlorococcus* concentrations focusing on data collected as part of the Atlantic Meridional Transect[62] and Pacific focused data from transects carrying the SeaFlow instrument (1897584 measurements across 33 cruises)[63]. Many cruises record very large shifts or transitions in *Prochlorococcus* abundance occurring on a scale of about 150 km (see Supplementary Fig. 3), with the North Pacific cruise MGL1704[64] containing two particularly obvious examples. Often, the *Prochlorococcus* abundance will change on the order of $10^5$ cells/mL in less than 150 km of distance, far exceeding any other variance along the cruise track. These events represent the cruise crossing a niche boundary from a region suitable for *Prochlorococcus* into one less suitable, or vice versa. The locations of these rapid shifts in abundance

were identified by finding the peaks in a Haar transform of the raw data (see Methods for more details)[65].

For each transition, we find coincident temperature and PAR (See Supplementary Fig. 1) using the Sea Surface Temperature[66] and MODIS Photosynthetically Available Radiation satellite derived datasets[67,68]. These temperature and PAR values are shown in Fig. 4 for all identified transitions which cross a concentration (75, 000 cells/ml) threshold taken as approximately half of peak values in the surface Pacific in the SeaFlow dataset (see Fig. 4a and Supplementary Fig. 1a–c). The collected transitions do not appear on a tight curve, and span a wide range of PAR and temperature values. This suggests that independent variation in temperature and PAR do not shift the spatial niche boundaries for *Prochlorococcus*. Returning to the Flombaum dataset (see Supplementary Fig. 4), a similar picture emerges when plotting surface data in temperature vs. PAR space. The overlap of observations greater than and less than the threshold estimates the ability of PAR and temperature to predict the threshold value. We find that both the transitions in Fig. 4c and the region of overlap in Supplementary Fig. 4 span most of the range of PAR observations and more than 15 degrees of temperature. These results are additionally corroborated by plotting the cruise track observations from SeaFlow and the AMT (See Supplementary Fig. 5). The threshold choice of 75,000 cells/mL sits at the base of the main data cloud which maintains a range of ~15 degrees (horizontal distance between solid black curves) independent of observed abundance.

The variability in the location of transitions in *Prochlorococcus* concentration shown in Fig. 4 does not appear strongly correlated with light and temperature. All transitions do, however, occur above a temperature of ~13 °C and the idea that there is a temperature threshold for *Prochlorococcus* growth is well established experimentally[41]. We thus use our observations of transitions to pose a simplified, two-state SDM for *Prochlorococcus* populations in the surface ocean that is consistent with experiments[41]. Similar nonlinear effects of temperature on general phytoplankton populations have also been observed[69]. Our two-state model predicts that *Prochlorococcus* concentrations can be expressed as a step-function in terms of temperature where *Prochlorococcus* concentrations are zero

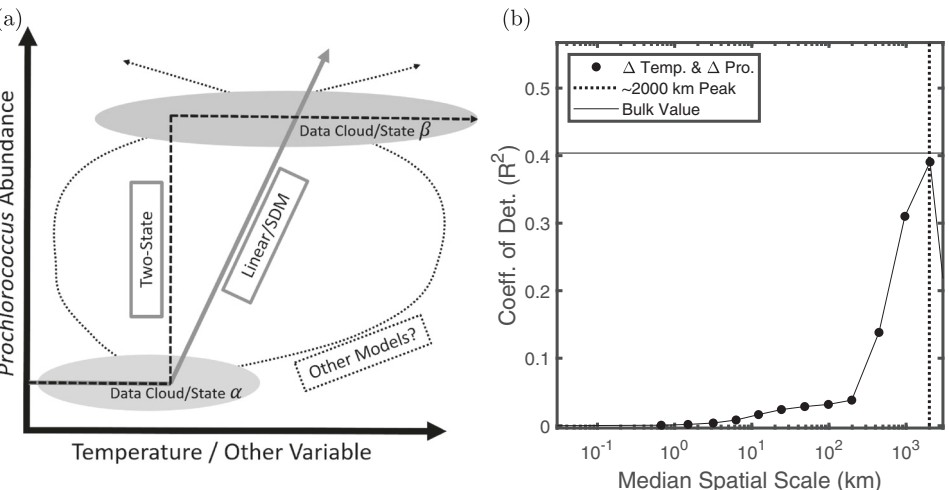

**Fig. 5 | Connecting data structure with predictability. a** A schematic showing how the spatial structuring of the data can be connected to the bi-modality of the data clouds. Two data clouds ($\alpha$ and $\beta$) are separated in both parameter and physical space. An infinite number of models, such as the finely dashed curves, fit these clouds equally well, but their predictions (slopes of curves) are divergent. The two state model is a step function (dashed curve) which predicts no changes with temperature at high abundance. A piece-wise linear model schematically approximating the SDM predicts large increases with changing temperature. **b** The coefficient of determination ($R^2$) between changes in *Prochlorococcus* abundance ($\Delta$ Pro.) and changes in temperature ($\Delta$ T) is plotted as a function of the spatial scale for the high resolution SeaFlow dataset[63].

(or set to $\log_{10} = 0$) below a certain temperature (13 °C in this case) and a constant above this temperature which is fit to the Flombaum dataset by minimizing the variance of the residuals. This idea is consistent with the Flombaum model as well as the ideas that went into forming it and can be viewed as a simplified version of the original model[14]. A schematic for how this model functions is shown in Fig. 5a.

We can compare the fit of this SDM to the full prediction of the Flombaum model in both logarithmic (the modified logarithmic space used to construct the original model[14]) and linear space (see Supplementary Fig. 9, and Supplementary Table 1). The $R^2$ values are highest for both models in log-space, and are very similar (0.44 and 0.415 for the original and two state models respectively), suggesting equivalence between the two models in the surface ocean. In Supplementary Fig. 9a, b the distributions of residuals in both linear and log-space are compared between the two models. The bi-modality of the residuals in log-space is matched by both models and the variance is equivalent between them (see Supplementary Table 1) suggesting that they have similar predictive power. In terms of variance, and consistent with Supplementary Fig. 9a, b, the Flombaum residuals have a variance of ~15% less than that of the two-state model in both linear and log-space. This model equivalence can be thought about in terms of the latitudinal prediction, first shown in Fig. 2c, as the dominant variation in the species' concentration occurs moving poleward. Residuals of the predictions of the two models are plotted in linear space, logarithmic differences are extremely small, as a function of latitude in Supplementary Fig. 9c. Noting that the maximal difference appears in the warm gyre and equatorial regions where abundances are normally high, we can reduce the dataset to these regions (between 30°S and 30°N) and gain some insight. In this limited portion of the range, $R^2 \approx 0.00$ for the two state model due to the warm temperatures being above the threshold whereas $R^2 \approx 0.04$ for the full model. In the tropics, the full model provides an ~4% reduction of the residual variance in linear space when compared to assuming a constant value (which essentially explains none of the variance in the tropics). The original model thus has minimal predictive power over changes inside the main range of *Prochlorococcus*.

We can use wavelets to test the effect of changing temperature on changes in *Prochlorococcus* abundance as a function of spatial scale. In Fig. 5b we explore the correlation between changes in *Prochlorococcus*

abundance and changes in temperature measured as a function of spatial distance. Operationally, this is done by convolving the SeaFlow dataset[63] with the normalized Haar wavelet and taking the correlation between the two convolutions. A flat and high $R^2$ curve would suggest that temperature has predictive power across spatial scales. However, the high $R^2$ values associated with the bulk dataset (and the Flombaum model) are only reached at large spatial scales. The continuous ramp in $R^2$ from 200-2000 km is caused as the convolution spreads information from the sharp transitions across larger and larger spatial scales (see Supplementary Fig. 10). This type of scale based analysis can be done with any model to determine if its power persists across a spectrum of spatial scales, or is caused by transitions between distinct regions.

## Discussion
Moving forward, we believe that the best predictions for the distribution of planktonic species like *Prochlorococcus* will eventually come from models which formally integrate both statistical and dynamical approaches. This combination has revolutionized weather forecasting, and should transform species prediction in the sea. This work takes a step in that direction by building an understanding of the differing predictions of dynamical and statistical models for *Prochlorococcus*. Observational data at different spatial-temporal scales can be used in an analogous fashion to laboratory experiments for testing the ability of organisms to grow and compete under different conditions. One of the promises of machine learning methods is that they can start with all of the data and fit a model which accurately balances the effects of changes across these varying spatial-temporal scales. Independent of how the model is produced, however, its efficacy can be independently tested against the separate scales used to construct it. As shown here, these tests can be quite simple. For marine plankton the spatial-temporal scales of variability can be quite distinct, spanning daily to monthly fluctuations in concentration to latitudinal shifts from crossing niche boundaries. The more scales a set of driving variables is predictive at, the more likely it will be predictive in new environments and in a changing climate.

Here, we focused on an SDM for *Prochlorococcus*[14], demonstrating that the model and its dependent variables (light and temperature) do not appear to maintain predictive power across both monthly

fluctuations in concentration and fluctuations in the spatial-temporal location of the spatial transitions. Where, then, does the Flombaum model attain its predictive power at the global scale? The majority of this model's predictive power in the surface ocean seems to come from the large change in population between places where *Prochlorococcus* is favored and places where it is not. This can be expressed by a two-state model which incorporates the idea of a thermal viability temperature, at a minimal cost of ~15% in the variance of the residuals. In terms of $R^2$, both models perform equally in log-space. In linear space, the original model performs marginally better, but when focusing on the main latitudinal range of the species, neither model does well. The $R^2$ of the original model drops to ~0.04 and for the two-state model $R^2 \approx 0$ as the temperature is higher than the threshold in this region. Considering that the observed, sharp transition in *Prochlorococcus* abundance occurs across >15 °C in temperature, the Flombaum model's predictions for the range increase in this species in a warming world is best interpreted as an estimate for the increase in its maximally viable range. The actual range may often be set by other drivers[12,70,71]. In places where *Prochlorococcus* is abundant, predictions for changes in *Prochlorococcus* concentration by the Flombaum model do not appear well supported.

These results can be put into context with other efforts to gain a more complete understanding of what sets the abundance patterns of *Prochlorococcus*. From a mechanistic perspective, there are a plethora of both top-down and bottom-up processes which can set the abundance of the species. Bottom-up factors like nutrients, temperature and light can directly influence the growth rates of the species which, for example, grows much slower at lower temperatures[41,72–74]. These slower growth rates provide a mechanistic rationale for including temperature in statistical models. Top-down controls are also important, with researchers implicating both grazer based[12,70] and viral[71] mechanisms to explain population shifts along transects in the North Pacific. Time is also an important factor[72]. The seasonal cycle forces large spatial oscillations in the boundaries of ecological regions in the ocean[57] and the poleward range of *Prochlorococcus* undergoes large (-10 degree) observed latitudinal changes over the season[70,75]. The temporal dynamics of sharp spatial transitions are likely one reason for the 15 degree temperature spread we observe in their location. Together, these results suggest that temperature sets the maximal range of *Prochlorococcus* populations, but that the actual range is often set by additional processes.

In terms of the surface populations of *Prochlorococcus*, our results suggest that the statistical power of the Flombaum SDM is generated by the large separation in parameter space between distinct population states. These states exist in colder nutrient rich waters with low *Prochlorococcus* abundances, and warmer nutrient poor waters with high abundances. As the ocean warms and becomes more stratified, waters are pushed from the cold, low abundance state to the warm, high abundance state. This generates the range expansion predicted both by the SDM and dynamical models. All model types agree that the range of *Prochlorococcus* will increase in a warming world, providing additional support for this prediction. However, predicted increases in abundance within the warm, low-nutrient, regime[14,15] appear hard to justify. We are left with the working hypothesis put forth by some statistical models[27] and by dynamical models[10] that concentrations of *Prochlorococcus* will decrease in the gyres as the planet warms. Certainly, complex feedbacks between temperature and nutrient cycles could lead to different predictions[59] but further work is required. The prediction of decreasing abundance inside the species' range should be tested with further experimental and modeling efforts. However, there is no evidence that the population will increase.

Machine learning methods and models are set to revolutionize our ability to predict the evolution of plankton communities by incorporating the effects of a high diversity of sparse observations.

Critical in this development is a parallel effort to simply and effectively test their predictions. Differences between model predictions and measurements can be thought about in terms of 'pattern errors' and 'magnitude errors'. Here, we demonstrate the importance of effectively splitting errors between their 'pattern' and 'magnitude' components as they contain different information. For *Prochlorococcus*, this was straightforward as a two-state, pattern only model fit the data well. We were thus able to conclude that the Flombaum model predicts range, but not concentration, and harmonize the predictions of current statistical and dynamical models for this species. Not all plankton prediction problems are this straightforward. Our conclusions were backed by a time series analysis, an analysis of the predictability of sharp spatial transitions, and a calculation as to the correlation structure of changes in *Prochlorococcus* and changes in temperature as a function of spatial scale. If temperature had maintained predictive power across spatial-temporal scales, we would have strong evidence that increasing temperature would lead to an increase in concentrations. For *Prochlorococcus*, this was not the case. However, we are hopeful that testing SDMs across spatial-temporal scales in this way will help find the models which are predictive in a changing sea. We suggest that models of this type need to demonstrate predictive power not only in distinct ocean basins, but across multiple distinct spatial-temporal scales before being extended to new environments and into a future climate.

## Methods

### Datasets

Our analysis included four datasets: the Flombaum dataset (the original dataset from which the Flombaum model was created[14]), the Hawaii Ocean Time-series (HOT)[48,60], the Atlantic Meridional Transect[62,76], and the SeaFlow dataset[63]. To simplify the analysis, we only included data taken near the sea surface, with a depth of at most 50 meters. No other measurements were excluded from the datasets.

The Flombaum, Atlantic Meridional Transect, and SeaFlow datasets were downloaded from the Simons CMAP project using the pycmap API (https://simonscmap.com/). The HOT dataset was downloaded from Hawaii Ocean Time-series Data Organization & Graphical System (data from http://hahana.soest.hawaii.edu/hot/hot-dogs/).

The measurements of *Prochlorococcus* abundance were colocalized with temperature and PAR measurements from datasets provided by CMAP. The temperature dataset was the GHRSST Level 4 AVHRR_OI Global Blended Sea Surface Temperature Analysis (GDS version 2) from NCEI, and the PAR dataset was the MODIS PAR dataset. Each *Prochlorococcus* measurement in the Flombaum dataset was associated with the nearest temperature and PAR measurement made on the same day. Temperature was colocalized to within ± 0.25° (28 km), and PAR was colocalized to within 9 km. Some PAR measurements were not available on certain days; those measurements were not used. The HOT dataset included temperature and PAR data, so no colocalization was necessary. Following the methods used by Flombaum, we accounted for the attenuation of light in water using the K490 attenuation coefficient. In the regions covered by the Flombaum dataset, we used a constant PAR attenuation coefficient[77] of k = 0.1 m⁻¹. As our analysis focused on the surface ocean, this attenuation did not make a significant difference in any of our results. For the scaling analysis of SeaFlow data, temperature and abundance were downloaded directly from the links included in[63].

### Time series analysis

To compute the direct correlations between temperature, PAR, and *Prochlorococcus* in HOT, we computed the average value of each variable (<50 meters depth) over each cruise (although many cruises only took one measurement). Each cruise was identified by an ID number in the HOT database, which allowed linking of various

measurements taken during the same cruise. The variance between measurements taken during the same cruise suggest a relative uncertainty of <1% in the measured temperature and PAR, and about 10% in the measured *Prochlorococcus* concentration. We treated data from each cruise (data taken over a few days) as individual data points.

To find the correlation between shifts in temperature, PAR, and *Prochlorococcus*, we sorted the cruises into bins based on the month in which they occurred. For each month represented in the dataset, we averaged the mean temperature, PAR, and *Prochlorococcus* over each cruise in that month. For each pair of consecutive months that were both represented among the cruises (there were several months in which no cruises occurred), we computed the differences in the average values of temperature, PAR, and *Prochlorococcus*. After applying these criteria, there were 123 pairs of consecutive months, which are represented in Fig. 3. The colocalization scheme is illustrated in Fig. 1d–f.

### Finding transitions using wavelets
We were particularly interested in locations where the population of *Prochlorococcus* abruptly changed, and sustained this change. To do this, we took the datapoints along a cruise and linearly interpolated them to form a continuous function $f$ (of *Prochlorococcus* population as a function of distance). We then convolved $f$ with the Haar function, defined below:

$$H_\alpha(t) = \begin{cases} 0 & t < -\alpha, \\ -1 & -\alpha \le t < 0, \\ 1 & 0 \le t < \alpha, \\ 0 & t \ge \alpha. \end{cases} \quad (1)$$

The convolution $H_\alpha * f$ measures the change in $f$ sustained over the interval $[t - \alpha, t + \alpha]$. By testing the number of peaks over each cruise as $\alpha$ varied, we found that the number of peaks sharply fell as $\alpha$ increased from 0, but began to stabilize before $\alpha = 150$ km. A lower value of $\alpha$ would detect more transitions, but these would be less significant; a greater value of $\alpha$ on the other hand may not distinguish two distinct transitions. Several examples are given in Supplementary Fig. 6. Transitions are seen as peaks and valleys as a function of both the wavelet width $\alpha$ and the distance along a cruise. Large stable transitions are seen as the peaks which persist independent of the size of the wavelet. Crucially, the location of these transitions is not sensitive to the choice of $\alpha$ as seen by the vertical stripes in Supplementary Fig. 6.

We therefore took $\alpha = 150$ km to be the standard wavelet for detecting transitions within each cruise and applied a low-level filter with threshold $C = 10$ cells/mL/km to remove small peaks. We considered a local minimum/maximum at $t$ to represent a transition if $|H_\alpha * f(t)| \ge C$, and the spatial distance between transitions was >100 km. This analysis yielded 31 transitions across 41 cruises. Using the time and geographical location of the transitions, we colocalized the set of transitions with temperature and PAR using the GHRSST and MODIS PAR datasets using the built in Python libraries in Simons CMAP (API from http://www.simonscmap.com).

### Building the two-state model
The two-state model was constructed in direct comparison with the Flombaum model, to be tested on the Flombaum dataset. To ignore effects from depth on variables such as PAR and temperature, we removed all data points with depth of >50 meters. In order to compare the results directly, we filtered the remaining data points by only using those which had temperature, PAR, and *Prochlorococcus* measurements. The two-state model was then constructed to return a constant value $C$ if the temperature $T \ge 13$, and 0 if $T < 13$. We chose $C$ to minimize the variance of the residuals, when comparing the results from the two-state model and the measured population of *Prochlorococcus*

in the Flombaum dataset. We found $C \approx 42000$ cells/mL so that

$$\mathcal{C}(T) = \begin{cases} 42000 & T \ge 13, \\ 0 & T < 13. \end{cases} \quad (2)$$

### Reporting summary
Further information on research design is available in the Nature Portfolio Reporting Summary linked to this article.

## Data availability
All data used in this study is publicly available through the Simons Foundation CMAP (http://www.simonscmap.com, pycmap API available at https://github.com/simonscmap/pycmap/archive/master.zip), the listed resources in the Methods section and the Supplementary Information. Data downloadable from the Simons CMAP project using the pycmap API include: the Flombaum dataset (the original dataset from which the Flombaum model was created[14]); the Atlantic Meridional Transect[62,76]; the SeaFlow dataset[63]; the GHRSST Level 4 AVHRR_OI Global Blended Sea Surface Temperature Analysis (GDS version 2) from NCEI[78], and the PAR dataset (MODIS PAR dataset[68]). The full SeaFlow abundance and temperature dataset external to CMAP is https://doi.org/10.5281/zenodo.3994953, direct download link from zenodo https://zenodo.org/record/3994953/files/SeaFlow_allstats_v.13_2020-08-21.zip?download=1. The HOT dataset was downloaded from Hawaii Ocean Time-series Data Organization & Graphical System (data from http://hahana.soest.hawaii.edu/hot/hot-dogs/).

## Code availability
Code central to the manuscript can be found as part of the Supplementary Information as Supplementary Code.

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

## Acknowledgements

The authors would like to thank Pedro Flombaum and Adam Martiny for graciously assisting with model code. Additionally, Adam Martiny provided extensive comments which greatly improved the manuscript. Funding for this project was provided by the MIT UROP office (V.B. and M.C.), and by the Simons Foundation (553242 and 827829, C.L.F.).

## Author contributions

C.L.F designed project. V.B., M.C., and C.L.F designed and conducted analysis. C.L.F., V.B., and M.C. wrote the manuscript.

## Competing interests

The authors declare no competing interests.
