## [Peer Review File · Nature Communications]

Understanding Opposing Predictions of Prochlorococcus in a Changing ClimateREVIEWER COMMENTS

Reviewer #1 (Remarks to the Author):

The use of machine learning algorithms to understand biogeographical patterns in the ocean and predict future changes is becoming more and more widespread. Often these statistical based predictions are inconsistent with mechanistic based model predictions. Bain and co-authors present an interesting analysis of one statistical model that has been used to predict the abundance of prochlorococcus, an ecologically important phytoplankton. The analysis is robust and in general the paper is well written. My primary concern is that the paper (as currently written) is fairly limited in scope and doesn't make a strong case for the broader applicability of the study that would warrant publication in Nature Communications. For example, one of the primary conclusions stated in the paper is that they are able to reconcile the statistical model predictions with contradictory mechanistic model predictions – this would be of great value for a broad scientific audience. However, I do not feel that this point is supported by the analyses presented in the paper.

The paper nicely illustrates fundamental flaws in the Flombaum model which uses temperature and PAR to estimate Prochlorococcus abundance. Bain and co-authors demonstrate that the data is bi-modal and that the statistical model only has power in predicting presence versus absence and very little power in predicting actual abundance. Not surprising, the authors show that there is little relationship between variance in temperature or PAR alone and pro abundance. However, the broader implications of this case study and larger lessons learned were not apparent.

I found the most insightful aspect of the paper was the boundary location analysis and the demonstration that these boundaries are not well defined using temperature or PAR. I was left wondering how we could better predict these transitions? How well does Flombaum's model predict the transitions? Are there additional parameters which could add information?

It wasn't entirely clear to me how much new knowledge was gained with the two-state "mechanistic" model. (I find this name a bit misleading as it isn't mechanistic but rather a simplification aimed at understanding the original model).

Minor comments:

- line 69 which model are you referring to
- line 70 which authors?
- line 87-88 I do not follow the statement of how a statistical model 'may function as a mechanistic model'
- line 103 how are these opposing predictions reconciled
- line 166 I believe should reference figure 3c not 3b
- line 270-271 'the species concentration where it is abundant do not seem' → needs editing
- line 275 suggest changing to "We suggest that models of this type need to demonstrate ..."

Reviewer #2 (Remarks to the Author):

The authors examine one model of Prochlorococcus abundance, based on a large data set, published by Flombaum et al. in 2013; then they apply a Species distributions model (SDM) to the Flombaum data set and 3 more data sets of Prochlorococcus abundance. They identify the conflict between the predictions of Flombaum (predicting an increase in Prochlorococcus and picoplankton abundance with warming) and the observations and predictions of declining phytoplankton and picoplankton in the subtropical gyres.

The authors focus on correlations between *Prochlorococcus* populations in the surface ocean, light, and temperature under three situations: global surface data and the predictive power of the Flombaum parametric regression model; the correlations of light and temperature over time; and the spatial-temporal transitions between regions of high and low population levels.

The manuscript is addressing a relevant aspect that is been largely bypassed by scientists in the topic: the inconsistency between the predictions by Flombaum and predictions based on other models of climate warming. I agree with the authors that the model of Flombaum is close to a mechanistic model; and lack the environmental complexity and interactions required for ocean warming predictions.

A major lack of the present manuscript is the lack of references to the relevant literature on the topic. Besides Flombaum et al paper, there are several models on picoplankton and specifically on *Prochlorococcus* that give different predictions in the abundance of these small organisms with warming. I think this is an important lack, that reduce the quality of the manuscript and limits the discussion. A revised version must include those specific references and the discussion of the conclusions of other models published, even if they contradict Flombaum conclusions.

Although the description of the model applied and the results obtained are clearly described, it is difficult to find the author's conclusion. Also, the discussion must be extended to include other statistical models showing different results than those of Flombaum et al.

The title reads: "Reconciling Opposing Predictions of *Prochlorococcus* in a Changing Climate " How the present manuscript reconciles the opposing predictions must be explained more clearly in a revised manuscript and should include the description of the opposing predictions.

Specific comments

Abstract: "This result reconciles recent work demonstrating that niche and global computational models for *Prochlorococcus* predict opposing trends as the ocean warms."
"

It is timely that the paper analyzed this aspect but the results of the global computational models for *Prochlorococcus* are not described in the manuscript. The present manuscript lacks references and descriptions of other publications on *Prochlorococcus* responses to climate change.

Lines 35 -61 The introduction here is too general and unprecise. Needs some improvement. The authors refer to microbes, which is a very, very broad group. Then they pass to species distributions models, with most of the models corresponding to plants and animals, but not to microbes. T there is an abrupt transition from those general aspects to *Prochlorococcus*.

Lines 76-78. Correct, as the models cited predict a large increase in *Prochlorococcus* abundance

Lines 78–80. The sentence must be modified as the "predictions" of Flombaum were not used in many applications; it is the picocyanobacteria distribution data reported in Flombaum et al. which is used.

Lines 82- 83. Too general as none of those references includes a model/prediction specific for *Prochlorococcus*, and only one includes pico-sized phytoplankton. This must be improved as it is an important lack throughout the manuscript. Relevant references must be cited: Chus et al. 2014 doi: 10.1111/gcb.12562; Agusti et al. 2019 doi:

10.3389/fmars.2018.00506, which is including models for Prochlorococcus abundance in the subtropical gyres; Acevedo-Trejos et al. 2014, <https://doi.org/10.3389/fmars.2014.00015>

Lines 98-103 "The statistical power of the Flombaum model thus appears to be driven primarily from Prochlorococcus' inability to grow at the lowest temperatures, and its relatively constant abundance in the heart of its spatial range. The opposing predictions of the statistical and global climate models for Prochlorococcus appear reconciled. "

This is a constant idea through the manuscript that must be improved: reconciled, in what sense? It is not clearly explained.

Lines 104-106: "The current statistical models seem limited in their predictive power to determine changes in this population when it is already high, but well suited to predict the maximal extent of the species' range."

"current statistical models" is restricted in the present manuscript to the model by Flombaum et al. The manuscript obviates other statistical models with different predictions; for example, the Global Change model described in Agusti et al. 2019, was constructed using a large data set of Prochlorococcus and picophytoplankton and environmental parameters from the subtropical gyres. Their global change model predicts a decrease of Prochlorococcus with warming at the surface ocean, but an increase in waters below the surface layer.

Lines 138 -139. was HOTs data included in the study of Flombaum? And the AMT data? if this is the case, the sentence needs a better description.

Lines 199- The identification of the transitions is an original and interesting aspect. The extended Figure 4 needs some modification; there are light blue, dark blue and light and dark red, but the legend indicated " Red denotes that the sample contained less than 75, 000 Prochlorococcus cells/mL and blue more than 75, 000 cells/mL."

Lines 225 – 230, related to these results, there is a recent paper by Feng et al. 2021 (<https://doi.org/10.1029/2020GB006808>) identifying a nonlinear temperature effect on global ocean chlorophyll-a, where Chlorophyll-a increased with increasing temperatures up to around 14 °C and Chlorophyll-a decreased with increasing temperatures above 14 °C.

Lines 243-244 and below in implications. Variability and the niche are also dependent on nutrients availability, and species interactions as competition, and grazers. Other models when considering such parameters predict a decline in pico-sized populations (Chus et al. 2014; Acevedo-Trejos et al. 2014).

Summary Response to Reviewers: We appreciate the opportunity to revise our manuscript and believe that we have successfully addressed the criticisms of the reviewers. The paper is substantially improved. Both reviewers believe that reconciling the opposing predictions made by dynamical models and the Flombaum SDM is a major result. Both reviewers felt we needed to do a better job connecting our technical calculations to our conclusions. We have taken special care to re-write substantial portions of the manuscript in order to make this case more clearly. The Abstract, Introduction, and Discussion sections have been substantially re-written with extensive additional referencing. Additionally, we have included a continuous scale analysis to show exactly how the correlations between temperature and abundance found in bulk data emerge from the integration of sharp transitions across very large spatial scales. We are hopeful that this new analysis, along with substantial added text is able to clarify our main points. We show that the Flombaum model fails in its predictions for abundances in locations where *Prochlorococcus* is already high, nullifying the parts of its predictions which contradict dynamical models. This provides a parsimonious prediction for *Prochlorococcus* in a warming ocean where the spatial range increases and populations inside the heart of that range decrease. Our scale based approach can be applied generally to SDMs built for climate prediction, testing predictive power by exploring functionality across a spectrum of spatial-temporal scales. We hope the reviewers are satisfied with our changes. Their comments have made for a much better manuscript.

REVIEWER COMMENTS

Reviewer #1 (Remarks to the Author):

The use of machine learning algorithms to understand biogeographical patterns in the ocean and predict future changes is becoming more and more widespread. Often these statistical based predictions are inconsistent with mechanistic based model predictions. Bain and co-authors present an interesting analysis of one statistical model that has been used to predict the abundance of prochlorococcus, an ecologically important phytoplankton. The analysis is robust and in general the paper is well written. My primary concern is that the paper (as currently written) is fairly limited in scope and doesn't make a strong case for the broader applicability of the study that would warrant publication in Nature Communications. For example, one of the primary conclusions stated in the paper is that they are able to reconcile the statistical model predictions with contradictory mechanistic model predictions — this would be of great value for a broad scientific audience. However, I do not feel that this point is supported by the analyses presented in the paper.

Response: We apologize for the lack of clarity in our previous revision and hope that our extensive changes alleviate this issue. We agree that one of the manuscript's primary conclusions is that contradictory predictions of statistical vs. mechanistic models are reconciled in this case. We also agree

that this type of conclusion is what warrants publication in *Nature Communications*, and apologize that the connection between our technical results and this conclusion is unclear. In our paper, we first demonstrate that the Flombaum model derives its statistical power from the large separation in parameter space between regions where *Prochlorococcus* exists and where it does not. We additionally find no evidence in local fluctuations for causality between temperature or light and changes in the species' abundance. The two-component model we build is important for our argument because it explicitly derives all of its statistical power from the species range. Inside the species range, the abundance of *Prochlorococcus* is assumed to be a constant. Since the two-component model is as good as the full model, it suggests that changes in the mean abundance inside the species range are not well predicted by changes in temperature and light. Thus, outside of changes in the species spatial range, which both population dynamics models and the Flombaum model agree on, the Flombaum model has no predictive power. This is why the results are reconciled. Our re-written manuscript makes these points much more clearly starting in the abstract where we now write:

“A two-state model based on a single temperature threshold matches the original SDM in the surface ocean further confirming that its power is driven by discrete shifts. We conclude that the original SDM has little power to predict changes in regions where *Prochlorococcus* is already abundant. This analysis both resolves an important conflict in our understanding of *Prochlorococcus* populations and suggests that SDMs prove efficacy across a continuous spectrum of spatial-temporal scales before being trusted in a changing ocean.”

As the reviewer believes that our analysis is robust, we are hopeful that the revised manuscript will better explain this connection.

The paper nicely illustrates fundamental flaws in the Flombaum model which uses temperature and PAR to estimate *Prochlorococcus* abundance. Bain and co-authors demonstrate that the data is bi-modal and that the statistical model only has power in predicting presence versus absence and very little power in predicting actual abundance. Not surprising, the authors show that there is little relationship between variance in temperature or PAR alone and pro abundance. However, the broader implications of this case study and larger lessons learned were not apparent.

Response: We believe that the manuscript makes two main contributions. First, by demonstrating this specific SDM has no predictive power in regions where *Prochlorococcus* populations are already high, it resolves the outstanding conflict between statistical predictions based on temperature and dynamical model predictions. The sign of this effect is of major importance for future carbon cycling and ecosystem structure. We are left with the parsimonious prediction that *Prochlorococcus*' range will increase, and that its population will decrease in regions where it is currently abundant.

The previous version of the manuscript did a poor job connecting the

case study of *Prochlorococcus* with broader issues of SDM prediction. To help alleviate this issue, we have included a continuous spatial scale analysis showing that correlations between temperature and this species only emerge at large scales and that this is due to the averaging over discrete ecosystem regimes. We suggest that SDMs used to predict abundance changes should demonstrate predictive power over a spectrum of spatial-temporal scales.

For *Prochlorococcus* specifically, we have added a paragraph in the discussion section which connects the results of our study to mechanistic ideas for what sets the population of this globally important species.

I found the most insightful aspect of the paper was the boundary location analysis and the demonstration that these boundaries are not well defined using temperature or PAR. I was left wondering how we could better predict these transitions? How well does Flombaum's model predict the transitions? Are there additional parameters which could add information?

Response: Understanding the mechanisms behind what sets the spatial transitions for *Prochlorococcus* is a fascinating question. Recent work has suggested that nutrient supplies mediated by apparent competition with heterotrophic bacteria may set transitions of *Prochlorococcus* in the Pacific ocean. Others, suggest that viral dynamics may play a central role. We now cite these papers in the main text and include a paragraph in the discussion section which connects different mechanistic ideas with our analysis. Our analysis of globally distributed transitions including the success of a two-state model suggests that temperature sets the maximally viable range for the species, but that other processes like top-down controls are often important in setting its observed range. This is laid out in the vastly changed discussion.

It wasn't entirely clear to me how much new knowledge was gained with the two-state "mechanistic" model. (I find this name a bit misleading as it isn't mechanistic but rather a simplification aimed at understanding the original model).

Response: We apologize, as stated earlier, we clearly did not do a sufficient job of connecting our technical analysis to the broader implications and conclusions of our work. In our mind, the two-state model includes a single piece of information, corroborated by laboratory experiments, that *Prochlorococcus* fails to grow at low temperatures. This two-state model by definition has no ability to predict changes in *Prochlorococcus* abundance once the population is in the 'abundant' state, yet it has a very similar statistical significance to the Flombaum model itself. Thus, both population dynamics models and the Flombaum model agree that the range of *Prochlorococcus* will expand under climate change as more regions of the ocean enter the 'abundant' regime. However, as the Flombaum model has no statistical power to predict changes within the 'abundant' state (the species' main range), this part of the conflict is nullified. We have added substantial text to the discussion section to make this clear. We have also removed the framing of

this as a 'mechanistic model'.

Additionally, we have moved the figure and table associated with the two-state model to the Supplementary Information and replaced it with a figure containing both a schematic of how the different models look in parameter space and a continuous scale analysis showing how correlations with temperature emerge only at large spatial scales when transects cross between ecosystem states.

Minor comments:

- line 69 which model are you referring to

Response: This line refers to the Flombaum 2013/2020 models. We now write: “The importance of both *Prochlorococcus* and the model constructed in [1] make it ideal for exploring the extendability of SDMs for plankton prediction under climate change.”

- line 70 which authors?

Response: We apologize for the confusion, but believe that the restructuring from the previous comment has alleviated this issue.

- line 87-88 I do not follow the statement of how a statistical model ‘may function as a mechanistic model’

Response: We originally framed our argument in terms of whether a statistical model functions as a mechanistic one, but see why this was confusing. We have re-framed the argument to focus on the predictive power of the statistical models. In this specific instance we have re-written it so it now says “... we posit that if the population is highly correlated with temperature and light across multiple spatial-temporal scales, then it may generate accurate predictions under future conditions. This could be due either to the direct, causal, relationships between temperature, light and the relative growth rates of the organisms, or due to hidden mechanisms which connect temperature and light to nutrient and physical dynamics...” . Our point is that a model which works must work for a reason. Either the variables in the model are causal, or they are correlated with something that is. This is especially important if one is to believe predictions for changes based on imposed shifts in the dependant variables like temperature.

- line 103 how are these opposing predictions reconciled

Response: We again apologize for the confusion. We have removed these conclusions from the end of the Introduction. While maintaining the conclusions of the manuscript, we have move away from the word 'reconciled' in case there is a difference in interpretation, starting with the title. As stated elsewhere, we show that the Flombaum SDM is unable to predict changes in places where *Prochlorococcus* abundance is already high, leaving the predictions of decreasing abundance within the current range. Both models agree that the range will expand. We have done extensive re-writing to try and make these points clear. We would like to point the reviewer specifically to a new paragraph in the Discussion section (references omitted below, but present in main text) which we feel lays out the conclusion nicely:

“In terms of the surface populations of *Prochlorococcus*, our results suggest that the statistical power of the Flombaum SDM is generated by the large separation in parameter space between distinct population states. These states exist in colder nutrient rich waters with low *Prochlorococcus* abundances, and warmer nutrient poor waters with high abundances. As the ocean warms and becomes more stratified, waters are pushed from the cold, low abundance, state to the warm, high abundance, state. This generates the range expansion predicted both by the SDM and dynamical models. Predicted increases in abundance within the warm, low-nutrient, regime appear hard to justify. We are left with the working hypothesis put forth by some statistical models and by dynamical models that concentrations of *Prochlorococcus* will decrease in the gyres as the planet warms. Certainly, complex feedbacks between temperature and nutrient cycles could lead to something different but further work seems required.”

- line 166 I believe should reference figure 3c not 3b

Response: Thank you. This has been changed.

- line 270-271 'the species concentration where it is abundant do not seem' —i needs editing

Response: We agree that this is confusing. The sentence now reads “In places where *Prochlorococcus* is abundant, predictions for changes in *Prochlorococcus* concentration do not seem well supported, bringing statistical and computational models out of conflict...”.

- line 275 suggest changing to “We suggest that models of this type need to demonstrate ...”

Response: Thank you. This has been done.

Reviewer #2 (Remarks to the Author):

The authors examine one model of *Prochlorococcus* abundance, based on a large data set, published by Flombaum et al. in 2013; then they apply a Species distributions model (SDM)

to the Flombaum data set and 3 more data sets of *Prochlorococcus* abundance. They identify the conflict between the predictions of Flombaum (predicting an increase in *Prochlorococcus* and picoplankton abundance with warming) and the observations and predictions of declining phytoplankton and picoplankton in the subtropical gyres.

The authors focus on correlations between *Prochlorococcus* populations in the surface ocean, light, and temperature under three situations: global surface data and the predictive power of the Flombaum parametric regression model; the correlations of light and temperature over time; and the spatial-temporal transitions between regions of high and low population levels.

The manuscript is addressing a relevant aspect that is been largely bypassed by scientists in the topic: the inconsistency between the predictions by Flombaum and predictions based on other models of climate warming. I agree with the authors that the model of Flombaum is close to a mechanistic model; and lack the environmental complexity and interactions required for ocean warming predictions.

Response: We appreciate that the reviewer sees the importance of addressing this specific case study of conflict between dynamical and statistical models. We hope that the extensive re-writing has alleviated the reviewer's concerns.

A major lack of the present manuscript is the lack of references to the relevant literature on the topic. Besides Flombaum et al paper, there are several models on picoplankton and specifically on *Prochlorococcus* that give different predictions in the abundance of these small organisms with warming. I think this is an important lack, that reduce the quality of the manuscript and limits the discussion. A revised version must include those specific references and the discussion of the conclusions of other models published, even if they contradict Flombaum conclusions.

Response: We appreciate the reviewer's concerns and have completely rebuilt large sections of the manuscript. Specifically, we now introduce the problem by discussing predictions from both dynamical and statistical models. We more thoroughly discuss and explain why dynamical models predict that *Prochlorococcus* will decrease in abundance within the gyres in a warming world. We include many more citations in the new manuscript (74 vs 42). In regards to the statistical models, we were focused on the results of Flombaum's 2013 and 2020 paper and apologize for not engaging with contrary statistical predictions. We now prominently cite (Augusti 2019) as demonstrating how the sign of the temperature effect can change depending on how the model is constructed. We write: "Additionally, recent statistical work on a dataset of *Prochlorococcus* collected from new transects isolated in the subtropics, suggests that the temperature sensitivity of SDMS changes sign depending on which ancillary variables are included in the analysis [27]".

Although the description of the model applied and the results obtained are clearly de-

scribed, it is difficult to find the author's conclusion. Also, the discussion must be extended to include other statistical models showing different results than those of Flombaum et al.

The title reads: "Reconciling Opposing Predictions of *Prochlorococcus* in a Changing Climate" How the present manuscript reconciles the opposing predictions must be explained more clearly in a revised manuscript and should include the description of the opposing predictions.

Response: We did a very poor job making our conclusions clear and appreciate this opportunity to clarify. We believe that the manuscript makes two main points which are hopefully straightforwardly described in the new text. First, using a series of scaling analyses we show that the Flombaum model gets its statistical power in the surface ocean from the large separation in parameter space between regions with high and low abundance and that it does a poor job capturing fluctuations in places where *Prochlorococcus* is already abundant. Thus, it has little ability to predict concentration changes in places where *Prochlorococcus* is already abundant. This resolves the conflict between the Flombaum SDM which predicts increases in abundance in gyre regions and dynamical models which suggest the opposite. The Flombaum SDM predictions are unsubstantiated, and we explain why. Both classes of model additionally agree that warming temperatures will increase the spatial range. We go on to argue that our approach is general with the help of a new analysis of the correlations between abundance and temperature as a function of spatial scale. We conclude that models which show predictive power across a spectrum of spatial-temporal scales are more likely predictive in a changing sea. We have de-emphasized the word 'reconcile', for example changing it to 'Understanding' in the title. We hope that our extensive re-writing and additional analysis satisfies the reviewer.

Specific comments

Abstract: "This result reconciles recent work demonstrating that niche and global computational models for *Prochlorococcus* predict opposing trends as the ocean warms. " It is timely that the paper analyzed this aspect but the results of the global computational models for *Prochlorococcus* are not described in the manuscript. The present manuscript lacks references and descriptions of other publications on *Prochlorococcus* responses to climate change.

Response: We appreciate that the reviewer understands the importance of this topic and the specific case study of *Prochlorococcus*. We have dramatically re-written the Introduction and Discussion to include both dynamical and statistical predictions with many additional citations. The reviewer's point makes tremendous sense and we have re-framed the manuscript because of it.

Lines 35 -61 The introduction here is too general and unprecise. Needs some improvement. The authors refer to microbes, which is a very, very broad group. Then they pass

to species distributions models, with most of the models corresponding to plants and animals, but not to microbes. There is an abrupt transition from those general aspects to *Prochlorococcus*.

Response: We agree with the reviewer and have completely re-written the Introduction of the paper to lay out the general predictions from dynamical and statistical SDM type models.

Lines 76-78. Correct, as the models cited predict a large increase in *Prochlorococcus* abundance

Response: We have modified the text as suggested. We have changed 'large, systematic changes in *Prochlorococcus*' to 'large systematic increases in *Prochlorococcus*'.

Lines 78–80. The sentence must be modified as the “predictions” of Flombaum were not used in many applications; it is the picocyanobacteria distribution data reported in Flombaum et al. which is used.

Response: We have re-written the sentence which now reads: “These predictions have large implications for topics ranging from understanding future changes in global microbial biodiversity [46, 47] to carbon sequestration driven by biological export out of the surface ocean [48–50].”

Lines 82- 83. Too general as none of those references includes a model/prediction specific for *Prochlorococcus*, and only one includes pico-sized phytoplankton. This must be improved as it is an important lack throughout the manuscript. Relevant references must be cited: Chus et al. 2014 doi: 10.1111/gcb.12562; Agusti et al. 2019 doi: 10.3389/fmars.2018.00506, which is including models for *Prochlorococcus* abundance in the subtropical gyres; Acevedo-Trejos et al. 2014, <https://doi.org/10.3389/fmars.2014.00015>

Response: We apologize for this, and our confusing introduction in general. The second paragraph of the Introduction is now a paragraph explaining how and why dynamical models predict decreases in the population of small cells in the surface gyres. We have included all of the suggested references in the new text.

Lines 98-103 “The statistical power of the Flombaum model thus appears to be driven primarily from *Prochlorococcus*' inability to grow at the lowest temperatures, and its relatively constant abundance in the heart of its spatial range. The opposing predictions of the statistical and global climate models for *Prochlorococcus* appear reconciled. “

This is a constant idea through the manuscript that must be improved: reconciled, in what sense? It is not clearly explained.

Response: We again apologize for being so unclear in our conclusions, which we hope are laid out in the above responses, and in the improved main text. In regards to this comment, we decided to remove the conclusions from the end of the introduction. Hopefully, everything is now more logically constructed.

Lines 104-106: “The current statistical models seem limited in their predictive power to determine changes in this population when it is already high, but well suited to predict the maximal extent of the species’ range.”

“current statistical models” is restricted in the present manuscript to the model by Flombaum et al. The manuscript obviates other statistical models with different predictions; for example, the Global Change model described in Agusti et al. 2019, was constructed using a large data set of *Prochlorococcus* and picophytoplankton and environmental parameters from the subtropical gyres. Their global change model predicts a decrease of *Prochlorococcus* with warming at the surface ocean, but an increase in waters below the surface layer.

Response: This line was removed. We now happily include reference to other statistical models that do not agree with Flombaum. We now cite Agusti et al. 2019 directly with statements like “Additionally, recent statistical work on a dataset of *Prochlorococcus* collected from new transects isolated in the subtropics, suggests that the temperature sensitivity of SDMS changes sign depending on which ancillary variables are included in the analysis...”

Lines 138 -139. was HOTs data included in the study of Flombaum? And the AMT data? if this is the case, the sentence needs a better description.

Response: Although there is data from the two regions, Flombaum’s original study did not include HOT or the AMT data so we left the original wording.

Lines 199- The identification of the transitions is an original and interesting aspect. The extended Figure 4 needs some modification; there are light blue, dark blue and light and dark red, but the legend indicated “ Red denotes that the sample contained less than 75, 000 *Prochlorococcus* cells/mL and blue more than 75, 000 cells/mL.”

Response: Thank you. We’ve fixed the coloring in the figure.

Lines 225 – 230, related to these results, there is a recent paper by Feng et al. 2021 (<https://doi.org/10.1029/2020GB006808>) identifying a nonlinear temperature effect on global ocean chlorophyll-a, where Chlorophyll-a increased with increasing temperatures up to around

14 °C and Chlorophyll-a decreased with increasing temperatures above 14 °C.

Response: Thank you. We've added a line referring to this paper: "Similar nonlinear effects of temperature on general phytoplankton populations have also been observed [66]"

Lines 243-244 and below in implications. Variability and the niche are also dependent on nutrients availability, and species interactions as competition, and grazers. Other models when considering such parameters predict a decline in pico-sized populations (Chus et al. 2014; Acevedo-Trejos et al. 2014).

Response: We completely agree with the reviewer and have included these references as well as an entire paragraph discussing how our results fit in with the mechanisms believed to set *Prochlorococcus* abundance.

REVIEWER COMMENTS

Reviewer #1 (Remarks to the Author):

I thank the authors for carefully considering the previous round of comments. I found the paper much improved especially the introduction and discussion. However, I do not feel the two issues I raised in my original review have been completely resolved:

Issue 1: Resolving the 'contradictory predictions'. The following points now come across very clearly in the manuscript:

- The Flombaum model performs well when assessed globally but breaks down when assessed at finer temporal and spatial scales

- The primary issue is the bimodal nature of *Prochlorococcus* distributions.

- The two-state model which explicitly captures this bimodality does as well as Flombaum.

However, the implications of these findings are still not clear. Specifically:

- The authors claim that the two-state model does better "in the main latitudinal range of the species". This was not clear to me from the results presented in the paper (specifically Supplemental Figure 9c). If there is evidence showing this, the authors need to present it in a much clearer fashion.

- Similarly, the authors state "In places where *Prochlorococcus* is abundant, predictions for changes in *Prochlorococcus* concentration by the Flombaum model do not appear well supported". I agree with this statement but they do not show the two-state model does better.

I think there is value to showing the two-state model does as well as Flombaum. I also believe that the two-state model does not necessarily have to do better than Flombaum but the authors imply that it does. What evidence is there that the two-state model will do better at predicting abundance changes? There is still a large amount of variance in *Prochlorococcus* abundances not captured by either model.

The reader is left with the feeling that Flombaum is inaccurate but we don't have a good alternative other than going with a dynamical model. What is the path forward? Is the main take away that SDMs shouldn't be used for predictions? I don't think this is what the authors intend.

These comments relate to my original comment about "reconciling the statistical model predictions with contradictory mechanistic model predictions". In the rebuttal, the authors state "We are left with the parsimonious prediction that *Prochlorococcus*' range will increase, and that its population will decrease in regions where it is currently abundant." While this work addresses the first part of this statement but I do not see where they show the second part.

Issue 2: While the current draft is greatly improved, it still lacks the bigger take aways. The authors nicely show that there are fundamental issues with the Flombaum model and that, while it might be useful for range expansion, it cannot be used to predict abundance shifts. However, what should be done to predict *prochlorococcus* distributions? And how does this study inform how we should approach SDMs in general. In the rebuttal they nicely state:

"Our scale based approach can be applied generally to SDMs built for climate prediction, testing predictive power by exploring functionality across a spectrum of spatial-temporal scales. "

This would be fantastic. But this does not come through in the manuscript.

Minor comments:

I really like Figure 5a. I find Figure 5b difficult to understand what the take-away is that I am looking for.

Line 48: typo, "regions where nutrientS are more plentiful"

Line 46: typo, "changes in light and temperature have A larger impact"

Line 69: typo "with the nonlinear nature of ecosystem ??? suggests" (need another word after ecosystem)

Line 83: what do you mean by 'sort systematically'? Cluster?

Line 94: "These authors" —> which authors?

Line 161: typo "cluster suggests exploring how"....

Reviewer #2 (Remarks to the Author):

The authors made a deep and appropriate revision addressing all the concerns identified and the revised manuscript is now much clear and strongly improved. The conclusion is now clearly identified and relevant. I recommend publication.

Summary Response to Reviewers: After the initial round of reviews we embarked on an extensive revision. We are pleased that reviewer # 2 recognizes both the extensiveness of our revision and the newfound clarity in our analysis and conclusions. Reviewer # 1 requires some additional discussion in order to move forward. Below, we respond to the well laid out inquiries by reviewer # 1 and hope that the additional text makes the connections between our technical results and conclusions clear.

REVIEWER COMMENTS

Reviewer #1 (Remarks to the Author):

I thank the authors for carefully considering the previous round of comments. I found the paper much improved especially the introduction and discussion. However, I do not feel the two issues I raised in my original review have been completely resolved:

Response: We appreciate the reviewer’s understanding as to the depth and effort undertaken in this last round of revision and are of course disappointed that the reviewer did not find them satisfactory. We are hopeful that the additional work we have put into the manuscript will resolve the two issues in a clear way.

Issue 1: Resolving the ‘contradictory predictions’. The following points now come across very clearly in the manuscript:

- The Flombaum model performs well when assessed globally but breaks down when assessed at finer temporal and spatial scales
 - The primary issue is the bimodal nature of *Prochlorococcus* distributions.
 - The two-state model which explicitly captures this bimodality does as well as Flombaum.
- However, the implications of these findings are still not clear.

Response: We appreciate the clarity with which the reviewer summarizes the above technical results. We feel that some additional discussion will help the reviewer understand the implications better. Certainly, the Flombaum model has a good fit to the raw data when collected at the global scale. In our manuscript, we try to explain in a simplified way why the Flombaum model has a good fit to data at the global scale. Depending on why the fit is good, we might decide to trust some aspects of a model’s predictions, but not others. We leverage the concepts of “pattern errors” and “magnitude errors” which plague model validation efforts in oceanography. In our case “pattern error” means: Do we predict the location of the high and low states of *Prochlorococcus* abundance correctly? The “magnitude error” asks: Can we predict the mean and variability inside the high and low abundance regions? When data that has strong spatial patterning is fit to a set of predictors both pattern and magnitude skill are merged in the total R^2 . We use the two-state model in this work because essentially all of its skill is in the pattern category. By showing the equivalence of the two-state “pattern only” model with the full model we make it clear that the

Flombaum model’s skill comes from the pattern. This provides confidence in the Flombaum model’s prediction of *Prochlorococcus*’ range expansion in a warming world. It also suggests that the prediction of increases within the “high concentration” state are not substantiated. This is nice, because now all model types agree! Hopefully, the additional and revised text (laid out below) in the Introduction, Results, and Discussion sections will help to better elucidate the implications of our work.

Specifically:

- The authors claim that the two-state model does better “in the main latitudinal range of the species”. This was not clear to me from the results presented in the paper (specifically Supplemental Figure 9c). If there is evidence showing this, the authors need to present it in a much clearer fashion.

- Similarly, the authors state “In places where *Prochlorococcus* is abundant, predictions for changes in *Prochlorococcus* concentration by the Flombaum model do not appear well supported”. I agree with this statement but they do not show the two-state model does better.

I think there is value to showing the two-state model does as well as Flombaum. I also believe that the two-state model does not necessarily have to do better than Flombaum but the authors imply that it does. What evidence is there that the two-state model will do better at predicting abundance changes? There is still a large amount of variance in *Prochlorococcus* abundances not captured by either model.

Response: We are relieved that the reviewer agrees with our analysis suggesting that the Two-State model performs ‘as well as Flombaum’. We do not intend to imply that the Two-State formulation does better than Flombaum and, based on the above comment, think that it might be due to the following sentence from the discussion: “In linear space, the original model performs marginally better, but when focusing on the main latitudinal range of the species, the R^2 of the original model drops to $\approx .04$.” Inside the latitudinal range of the species the two-component model has zero predictive power on fluctuations because the temperature is always above the ‘threshold/viability’ temperature. We have now changed this sentence to make it abundantly clear that neither model has predictive power inside the species range: “In linear space, the original model performs marginally better, but when focusing on the main latitudinal range of the species, neither model does well. The R^2 of the original model drops to $\approx .04$ and for the two-state model $R^2 \approx 0$ as the temperature is higher than the threshold in this region.”

For context, the entire paragraph now reads

“Here, we focused on an SDM for *Prochlorococcus* [1], demonstrating that the model and its dependent variables (light and temperature) do not appear to maintain predictive power across both monthly fluctuations in concentration and fluctuations in the spatial-temporal location of

the spatial transitions. Where, then, does the Flombaum model attain its predictive power at the global scale? The majority of this model’s predictive power in the surface ocean seems to come from the large change in population between places where *Prochlorococcus* is favored and places where it is not. This can be expressed by a two-state model which incorporates the idea of a thermal viability temperature, at a minimal cost of $\approx 15\%$ in the variance of the residuals. In terms of R^2 , both models perform equally in log-space. In linear space, the original model performs marginally better, but when focusing on the main latitudinal range of the species, neither model does well. The R^2 of the original model drops to $\approx .04$ and for the two-state model $R^2 \approx 0$ as the temperature is higher than the threshold in this region. Considering that the observed, sharp transition in *Prochlorococcus* abundance occurs across more than 15°C in temperature, the Flombaum model’s predictions for the range increase in this species in a warming world is best interpreted as an estimate for the increase in its maximally viable range. The actual range may often be set by other drivers [70, 71, 14]. In places where *Prochlorococcus* is abundant, predictions for changes in *Prochlorococcus* concentration by the Flombaum model do not appear well supported.”

It was not our intention to imply that the two-state model does better than Flombaum in “the main latitudinal range of the species”. We completely agree with the reviewer that neither model captures the variance of *Prochlorococcus* populations in this region. We hope that the above change reaffirms this. We agree with the reviewer that neither model is predictive in regions where *Prochlorococcus* populations are already high.

The reader is left with the feeling that Flombaum is inaccurate but we don’t have a good alternative other than going with a dynamical model. What is the path forward? Is the main take away that SDMs shouldn’t be used for predictions? I don’t think this is what the authors intend.

Response: We do not intend to suggest that dynamical models are the only path forward. Please see our first response to Issue #2 (below), as we feel these questions are directly related to that issue. We are pleased that our manuscript elicits these sort of questions.

These comments relate to my original comment about “reconciling the statistical model predictions with contradictory mechanistic model predictions”. In the rebuttal, the authors state “We are left with the parsimonious prediction that *Prochlorococcus*’ range will increase, and that its population will decrease in regions where it is currently abundant.” While this work addresses the first part of this statement but I do not see where they show the second part.

Response: We are a bit confused by this comment. A version of this sentence appears in the discussion section in the following paragraph:

“In terms of the surface populations of *Prochlorococcus*, our results suggest that the statistical power of the Flombaum SDM is generated by the large separation in parameter space between distinct population states. These states exist in colder nutrient rich waters with low *Prochlorococcus* abundances, and warmer nutrient poor waters with high abundances. As the ocean warms and becomes more stratified, waters are pushed from the cold, low abundance, state to the warm, high abundance, state. This generates the range expansion predicted both by the SDM and dynamical models. Predicted increases in abundance within the warm, low-nutrient, regime [1, 2] appear hard to justify. We are left with the working hypothesis put forth by some statistical models [27] and by dynamical models [12] that concentrations of *Prochlorococcus* will decrease in the gyres as the planet warms. Certainly, complex feedbacks between temperature and nutrient cycles could lead to something different [59] but further work seems required.”

Dynamical models (and some statistical ones) predict that the range will increase, but that the abundance inside that range will decrease. We believe that our work calls into question the validity of predictions for increased abundance inside the species range. A prior comment by the reviewer suggests that they are in agreement with this point. Thus, “We are left with the parsimonious prediction that *Prochlorococcus*’ range will increase, and that its population will decrease in regions where it is currently abundant.”

We have added the following line to the end of the paragraph: “The prediction of decreasing abundance inside the species’ range should be tested with further experimental and modeling efforts. However, there is no evidence that the population will increase.” Hopefully, this makes it clear that the prediction of decreasing abundances requires additional scrutiny.

Issue 2: While the current draft is greatly improved, it still lacks the bigger take aways. The authors nicely show that there are fundamental issues with the Flombaum model and that, while it might be useful for range expansion, it cannot be used to predict abundance shifts. However, what should be done to predict prochlorococcus distributions?

Response: Moving forward, we believe that the best predictions for the distribution of planktonic species like *Prochlorococcus* will come from models which formally integrate both statistical and dynamical approaches. We feel strongly that phytoplankton modeling and prediction is on the same path as weather modeling. In that field, dynamical and statistical approaches are combined to enhance the accuracy of forecasts. We now start the discussion with this point:

“Moving forward, we believe that the best predictions for the distribution of planktonic species like *Prochlorococcus* will eventually come from models which formally integrate both statistical and dynamical approaches. This combination has revolutionized weather forecasting, and should transform species prediction in the sea. This work takes a step in that direction by building an understanding of the differing predictions of dynamical and statistical models for *Prochlorococcus*.”

Both statistical and dynamical models are susceptible to the strong imprint of global circulation patterns. Fundamentally, any model which is built from global patterns must attempt to remove this overprinting if they want to predict how the system will evolve.

We have added the following paragraph to the Introduction to introduce this issue more clearly:

“Determining the validity of both these model types can be difficult because of the spatial patterning of ocean data [32]. The ocean can be separated into physical and biophysical provinces with sharp spatial transitions [33, 34]. This, combined with the nonlinear nature of ecosystem population dynamics, suggests distinct population regimes in the sea [35, 36, 37]. Differences between model predictions and measurements can thus be thought about in terms of ‘pattern errors’ and ‘magnitude errors’ [38]. Differences can be caused by the shifting of regime boundaries in space, or by the modification of population levels within a province itself [39]. When statistical models are built from global datasets, both pattern and magnitude errors can influence the goodness of fit. Thus, it becomes critical to understand why a model has a good fit in order to determine under which circumstances its predictions should be trusted.”

And how does this study inform how we should approach SDMs in general. In the rebuttal they nicely state:

“Our scale based approach can be applied generally to SDMs built for climate prediction, testing predictive power by exploring functionality across a spectrum of spatial-temporal scales. “

This would be fantastic. But this does not come through in the manuscript.

Response: One of the underlying technical points in the manuscript is that strong spatial structuring can lead to the formation of models whose high R^2 values do not suggest predictive power in a changing world. As discussed earlier, the fundamental issue is the interplay between “pattern” and “magnitude” errors. For *Prochlorococcus* this problem manifested through the two distinct high and low concentration states existing at low and high latitudes, respectively. In figure 5b we leverage the high-resolution transect data from SeaFlow to explicitly demonstrate this issue with a continuous

analysis. We use wavelets to show how changes in temperature and changes in *Prochlorococcus* abundance are correlated, but only at large spatial scales when sharp spatial transitions have been crossed. This issue generates the sharp peak in the plot of correlation vs. spatial scale.

This type of analysis could be done with any model, asking how the model predicts variability at different spatial scales. Ideally, the model has equal and large predictive power at all spatial scales. This type of behavior would add to confidence in the model's ability to predict magnitude shifts when predictors like temperature are changed. Although intended, we do not believe that the final paragraph in the results section made this clear. We have altered some of the confusing language and the paragraph now reads:

“We can use wavelets to test the effect of changing temperature on changes in *Prochlorococcus* abundance as a function of spatial scale. In figure 5b we plot the correlation between changes in *Prochlorococcus* abundance and changes in temperature measured as a function of spatial distance. Operationally, this is done by convolving the SeaFlow dataset [63] with the normalized Haar wavelet and taking the correlation between the two convolutions. A flat and high R^2 curve would suggest that temperature has predictive power across spatial scales. However, the high R^2 values associated with the bulk dataset (and the Flombaum model) are only reached at large spatial scales. The continuous ramp in R^2 from 200-2000 km is caused as the convolution spreads information from the sharp transitions across larger and larger spatial scales (see Supplemental Figure 10). This type of scale based analysis can be done with any model to determine if its power persists across a spectrum of spatial scales, or is caused by transitions between distinct regions.”

We do believe that this manuscript provides a path towards formally testing SDMs in a simplified way by testing their predictive power on time-series, when reduced to spatial transitions, and in a continuous fashion. If models retain predictive power in these situations we gain confidence that they will retain predictive power when the predictors are changed. In an attempt to make this clear, we have added the following section inside the final paragraph of the discussion section:

“Here, we demonstrate the importance of effectively splitting errors between their ‘pattern’ and ‘magnitude’ components as they contain different information. For *Prochlorococcus*, this was straightforward as a two-state, pattern only, model fit the data well. We were thus able to conclude that the Flombaum model predicts range, but not concentration, and harmonize the predictions of current statistical and dynamical models for this species. Not all plankton prediction problems are

this straightforward. Our conclusions were backed by a timeseries analysis; an analysis of the predictability of sharp spatial transitions; and a calculation as to the correlation structure of changes in *Prochlorococcus* and changes in temperature as a function of spatial scale. If temperature had maintained predictive power across spatial temporal scales, we would have strong evidence that increasing temperature would lead to an increase in concentrations. For *Prochlorococcus*, this was not the case. However, we are hopeful that testing SDMs across spatial-temporal scales in this way will help find the models which are predictive in a changing sea.”

We really appreciate the clear comments by the reviewer and hope that the additional text has made things more straightforward.

Minor comments: I really like Figure 5a. I find Figure 5b difficult to understand what the take-away is that I am looking for.

Response: We are glad that Figure 5a was clear, but are disappointed that Figure 5b is confusing. Please see the prior response for more detail.

Line 48: typo, “regions where nutrientS are more plentiful”

Response: Thank you, this has been fixed.

Line 46: typo, “changes in light and temperature have A larger impact”

Response: Thank you, this has been fixed.

Line 69: typo “with the nonlinear nature of ecosystem ??? suggests“ (need another word after ecosystem

Response: This now reads “combined with the nonlinear nature of ecosystem population dynamics, suggests”.

Line 83: what do you mean by ‘sort systematically’? Cluster?

Response: We have replaced ‘sort systematically’ with ‘cluster systematically’.

Line 94: “These authors” —¿ which authors?

Response: We have replaced ‘These authors’ with ‘Flombaum et al’.

Line 161: typo “cluster suggests exploring how”

Response: Thank you, we have fixed this typo.

Reviewer #2 (Remarks to the Author):

The authors made a deep and appropriate revision addressing all the concerns identified and the revised manuscript is now much clear and strongly improved. The conclusion is now clearly identified and relevant. I recommend publication.

Response: We are glad that the substantial revisions we have done to the original manuscript satisfied the reviewer. We appreciate the comments and feel that the current version is a much stronger contribution. Thank you for your time and efforts on the review.

REVIEWERS' COMMENTS

Reviewer #1 (Remarks to the Author):

I thank the authors for taking the time to carefully revise the manuscript to address the concerns I raised in my previous review. I particularly like the new framing of "pattern and magnitude errors". I feel this does a fantastic job setting up the problem and conveying clearly to the reader the challenges at hand. The rewrite of the Discussion has also significantly improved the 'take-away' messages for the paper and I believe will provide broader impact.

There are a couple of places where the language could be tightened (e.g. last sentence of the abstract). But otherwise, I am happy with the current version of the manuscript.